# Measuring the Quality of Life in Patients with Chronic Venous Disease before and Short Term after Surgical Treatment—A Comparison between Different Open Surgical Procedures

**DOI:** 10.3390/jcm11237171

**Published:** 2022-12-02

**Authors:** Sergiu-Ciprian Matei, Cristina Ștefania Dumitru, Daniela Radu

**Affiliations:** 1Department of Surgery, “Victor Babeș” University of Medicine and Pharmacy Timișoara, EftimieMurgu Sq. no. 2, 300041 Timișoara, Romania; 21st Surgical Clinic, “Pius Brînzeu” Emergency County Hospital, LiviuRebreanu Boulevard no. 156, 300723 Timișoara, Romania; 3Department of Microscopic Morphology/Histology, Angiogenesis Research Center, “Victor Babes” University of Medicine and Pharmacy, Sq. EftimieMurgu no. 2, 300041 Timișoara, Romania

**Keywords:** chronic venous disease, quality of life (QoL), CIVIQ-20, cryostripping, phlebectomies, venous stripping

## Abstract

Chronic venous disease (CVD) is a common pathology that significantly affects the quality of life (QoL) of patients. Methods: QoL was assessed in 317 patients diagnosed with CVD who underwent surgeries, including cryostripping (*n* = 113), high ligation and stripping (HL&S, *n* = 96), and phlebectomies (*n* = 108). CVD symptoms and QoL were assessed before surgery and 2 weeks after surgery using the following questionnaires: CIVIQ-20, VAS, Eq-5D, PHQ-9 and GAD-9. Results. The results reveal a significant correlation (*p* < 0.05) between CEAP score and QoL questionnaires performed preoperatively and postoperatively in all three surgical technique groups, with a statistical improvement postoperatively. Phlebectomy had the best postoperative QoL score (*r* = 0.495) compared to the other two types of procedures. Conclusions: Analyzing patients’ subjective perception following conventional surgery for CVD treatment, an improved QoL is observed both in functional and psychosocial aspects, even early postoperatively. Classical surgical procedures remain an effective and feasible option in CVD treatment.

## 1. Introduction

Chronic venous disease (CVD) is a frequent pathology and a common health care problem. The prevalence of this condition remains underestimated, early evidence of CVD being frequently overlooked by general practitioners [1]. Clinical manifestations of CVD can range from mild to severe, such as telangiectasia, varicose veins (most notably), lipodermatosclerosis or venous ulceration, and pain is one of the most common symptoms [2,3]. It is well-known that CVD negatively impacts patient’s quality of life (QoL), both at the physical and the psychological levels [4]. Patients initially seek treatment to relieve symptoms of leg pain, discomfort, heaviness, and swelling, and for an overall QoL improvement [5]. A significant percentage of patients with CVD require various surgical procedures as a therapeutic method, those having a significant psychosocial and economic impact [6,7]. In the recent decade, the recommendations for managing symptomatic varicose veins have changed dramatically due to the rise of minimally invasive endovascular techniques. However, in low- and middle-income countries, classic surgical procedures remain the first treatment option in state hospitals. The main surgical procedures used are phlebectomies and high ligation and stripping (HL&S) [8,9]. In addition, in some specialized centers, insufficient vein removal is also practiced by cryostripping [10].

Surgery outcomes can be objectively evaluated via clinical examination and color-flow duplex ultrasound, which is the gold standard in this regard [11,12]. However, QoL evaluation requires various questionnaires and involves a subjectivity grade from the patient. Different questionnaires proved to be reliable instruments in QoL evaluation. The 20-itemChronic Venous Disease quality-of-life Questionnaire (CIVIQ-20) is a valid score for the assessment of treatment effects in multinational studies [13], and the pain Visual Analog Scale (VAS) is a valid tool for measuring pain [14]. EuroQol-5 Dimension (EQ-5D) is an instrument which evaluates the generic quality of life developed in Europe and is widely used [15]. The Patient Health Questionnaire (PHQ-9) and Generalized Anxiety Disorder (GAD-7) scales are brief well-validated measures for detecting and monitoring depression and anxietydisorders associated with a decreased QoL [16]. Previous studies proved that about third of the patients with symptomatic varicose veins struggle with the burden of depression [17].

This study aims to evaluate the short-term impact of surgical treatment on the QoL in patients with CVD, analyzing their subjective perception regarding treatment outcomes and at the same time comparing the results between the different surgical open procedures.

## 2. Materials and Methods

### 2.1. Patients

This prospectivestudy included 317 patients which were admitted for surgical treatment in the Phlebology Department (1st Surgical Clinic, “Pius Brînzeu” Emergency County Hospital, Timișoara, Romania), between January 2019 and December 2021. All the patients were evaluated preoperatively by clinical examination and duplex ultrasound, and venous reflux was found. The patients were subsequently operated upon, with different open procedures being performed. Only venous segments with venous reflux were surgically excluded, according to the preoperative ultrasound mapping of the superficial venous network of the lower limb.Depending on the venous reflux site, the type of surgical intervention was chosen, taking into account the possible variants in our state healthcare system (procedures that are covered by health insurance). Patients were subdivided.Then, according to the type of surgery that was practiced, they were divided into three groups, as follows: cryostripping (*n* = 113), high ligation and stripping (HL&S, *n* = 96), and phlebectomies (*n* = 108).

### 2.2. Quality of Life Assessment

All patients received quality of life assessment questionnaires. Chronic venous disease symptoms and quality of life were assessed before surgery and two weeks after surgery using the following questionnaires: CIVIQ-20, VAS, Eq-5D, PHQ-9, and GAD-9.

### 2.3. Inclusion and Exclusion Criteria

Inclusion criteria for the study included correct and complete completion of QoL questionnaires by patients one day before surgery and two weeks after surgery and signing of informed consent. Patients who had undergone previous varicose vein surgery did not sign the informed consent or did not present themselves for the two weeks postoperative follow-up were excluded from the study, as were the patients diagnosed with other chronic diseases that may influence QoL(heart failure > NYHA II, atrial fibrillation, documented myocardial infarct, deep vein thrombosis, asthma, chronic obstructive pulmonary disease, complications of other chronic diseases) or that could have erroneously influenced the completion of the questionnaires (patients with documented psychiatric pathologies).

### 2.4. Data Analysis

Statistical analyses were completed using MedCalc^®^ Statistical Software version 20.015 (MedCalc Software Ltd., Ostend, Belgium; 2021). The results were statistically analyzed using the Wilcoxon signed rank test and Spearman test. The Shapiro–Wilk test was used toanalyze the normality distribution of variables. The resulting *p*-value < 0.05 was considered statistically significant.

## 3. Results

224 women and 93 men (mean age 51.6 ± 14; range 21–77 years) diagnosed with CVD were enrolled in the study and the immediate follow-up period was 2 weeks postoperatively. Of these, 113 patients were treated by cryostripping, 96 patients were treated by HL&S and 108 patients were treated by phlebectomies. Therefore, we conducted three study groups according to the type of surgery, where we analyzed preoperative and postoperative QoL at 2 weeks in all patients, also the Clinical-Etiological-Anatomical-Pathophysiological (CEAP) classification was used as a reference assessment. No difference was observed between the three groups in terms of the age of the patients and their gender. In terms of age groups, it was observed that the age group 41–55 years was predominant in the number of patients with a CEAP score 3 in all three surgical techniques (Table 1).

Results reveals a significant correlation (*p* < 0.05) between the CEAP score and the QoL questionnaires performed preoperatively and postoperatively in all three surgical technique groups (Table 2). However, to analyze the differences between the three groups, we used the Spearman rank correlation coefficient (rho = *r*) to search for strong associations in the patient data. According to the correlation coefficient *r*, the postoperative data are strongly significant, representing an improvement in quality of life (Figure 1). In the HL&S group, there was a predominantly weak association (*r* = 0.162–0.352); in the cryostripping group, there was a predominantly weak/moderate association (*r* = 0.281–0.615); and in the phlebectomy group, there was a moderate association (*r* = 0.235–0.641).To better appreciate the statistical difference, we averaged the preoperative and postoperative correlation coefficient *r* for each group of patients for the HL&S group preoperative *r* = 0.236, postoperative *r* = 0.272; cryostripping preoperative *r* = 0.371, postoperative *r* = 0.416; phlebectomy preoperative *r* = 0.385, postoperative *r* = 0.495.Thus, it follows that phlebectomy as a surgery had the best QoL score compared to the other two types of surgery.

The most noticeable patient improvement was seen postoperatively in the CIVIQ-20 score for both phlebectomy and cryostripping surgical techniques. The CIVIQ-20 score involved assessment of symptom involvement in usual daily activities, sleep impairment, various motility actions, and mood.

## 4. Discussion

Chronic venous disease is a highly prevalent condition in the general population, generating variable reasons for consultation that can alter the patient’s quality of life. Even if in some cases it can be asymptomatic, more frequently it causes subjective symptoms or lead to objective alterations, such as edema, cutaneous alterations, and venous leg ulcers. Varicose veins are the most common clinical manifestation. They are a progressive degenerative disease of the venous walls in the superficial venous system of the legs which can decisively impair the quality of life of those affected. The treatment of chronic venous diseases targets the improvement of the subjective complaints and objectively alterations [18,19]. From the clinical point of view, treatment goals for patients with CVD include reduction of edema and lipodermatosclerosis, and ulcers prevention or healing. Regarding venous ulceration, randomized trials have found that superficial venous surgery did not improve ulcer healing, but significantly reduced ulcer recurrence compared to compression therapy alone [20]. Venous leg ulcer is particularly associated with a significant decrease in QoL [21,22], with prompt treatment measures being essential in improving the status of these patients.

A wide range of therapeutic methods can be used to treat CVD. In general, invasive treatments (surgical and endoluminal) were superior to conservative management in eliminating varicose veins and decreasing ulcer recurrence rates [23]. Stripping operations and the less invasive endovenous thermal ablation show comparable results for saphenous vein varicose treatment [19]. However, in most countries with medium and low incomes, only classical procedures are practiced in state hospitals for economic reasons. This study included only patients operated by classical techniques: phlebectomies, HL&S, and cryostripping. The type of procedure was chosen in accordance with the topography and morphology of insufficient veins and venous reflux sites objectivated by duplex ultrasound.

QoL measures have become a vital and often required part of health outcomes appraisal. For populations with chronic disease, measurement of QoL provides a meaningful way to determine the impact of health care. Over the past 20 years, many instruments have been developed that purport to measure QoL [24], with EQ-5D being frequently used in this regard. Particularly for CVD, CIVIQ-20 proved to be a useful instrument for QoL measurement. Previous studies also revealed that CIVIQ-20 questionnaire results could be correlated with clinical signs and symptoms related with CVD [25], and the score results improve after the treatment, being directly correlated with the clinical evolution [26].

Pain is a constant symptom among CVD patients, and pain VAS could provide significant information regarding therapeutic outcomes. Because symptoms of anxiety and depression have been shown to significantly correlate with lower health-related QoL scores [27], questionnaires like PHQ-9 and GAD-9 may be also useful in evaluating patients’ perceptions regarding the outcomes. Even though all the questionnaires used in this study are well studied and valid instruments, their results are subject to a degree of subjectivity. 

Other questionnaires useful in venous disease QoL assessment were also described in previous studies. ABC-V (Assessment of Burden in Chronic diseased Venous), AVVQ (Aberdeen Varicose Vein Questionnaire) CCVUQ(Charing Cross Venous Ulceration Questionnaire), SPVU-5D (Sheffield Preference-based Venous leg Ulcer questionnaire with 5 Dimensions), SQOR-V (Specific Quality of life and Outcomes Responsed Venous), VEINES-QOL/Sym (VenousINsufficiency Epidemiological and Economic Study on Quality of Life/Symptoms) and VLU-QOL (Venous Leg Ulcer Quality Of Life questionnaire) are other examples of questionnaires related with CVD evaluation [28]. One decisive factor in choosing the questionnaires used in this study was the fact that all of them are translated into Romanian and validated.

QoL is an important outcome measure in the treatment for chronic venous disease [29]. This study results revealed a significant QoL improvement in all the study groups regardless of the used procedure, even if the follow-up was a short one to two weeks. The prognosis of CVD is greatly dependent upon the ability of the patients to optimize their health-related behaviors (mainly compliance to compression stockings, physical activity, and diet) [30] and on lifestyle changes in association with treatment leading to an improvement in QoL. However, a slightly better improvement seems to be observed in the phlebectomy group. This observation could be explained by the fact that phlebectomies are practiced through stab incisions and are associated with a shorter hospitalization period. Additionally, the patients which required phlebectomies presented venous reflux on collateral or perforating branches with apparently healthy saphenous veins, a fact that can be associated with milder symptoms. Our results are in accordance with data from the literature, supporting the idea that percutaneous phlebectomies should be considered the method of choice for side branch varicose vein removal; however, recurrences of varicose veins are frequent [19,31]. The main advantages of phlebectomies compared to the other classic procedures (HL&S or cryostripping) are the fact that they can be performed in an outpatient setting without the need for hospitalization [32,33].

A limitation of this study can be considered the fact that there was no long-term evaluation of the patients (a period of at least one year from the baseline). This is due on the one hand to the fact that once the symptoms have subsided, not all patients return for regular long-term consultations and on the other hand due to the fact that the study period has recently ended; we do not yet have long-term results for some of the patients. However, we are considering a future study in which we will compare the results of early and one-year postoperative questionnaires results and treatment impact on QoL, when we will collect long-term follow-up data for a larger number of patients.

## 5. Conclusions

Analyzing patients’ subjective perception following conventional surgery for CVD treatment, an improved QoL is observed both in functional and psychosocial aspects, even early postoperatively. Along with the remission of the symptoms, there is also a remission of other disorders associated with CVD, such as nervousness, anxiety, tension, or depression. Classical surgical procedures remain an effective, safe, and feasible option in CVD treatment, especially in state hospitals in low- or mid-income countries where endovenous procedures are not available. From that kind of procedures, phlebectomies seem to represent the optimal treatment option in terms of QoL according to both physicians, and patients’ subjective appreciation.

## Figures and Tables

**Figure 1 jcm-11-07171-f001:**
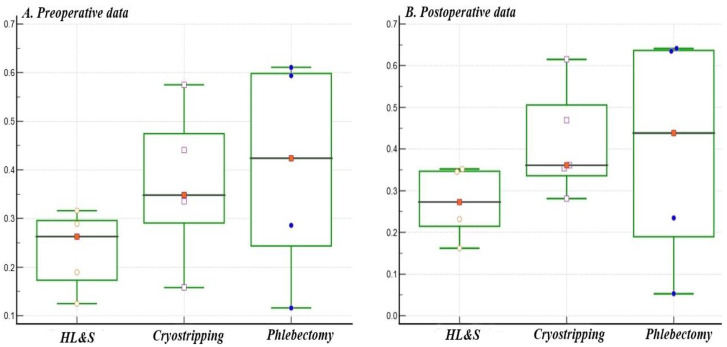
Data of the three groups compared preoperative (**A**) and postoperative (**B**). A graphical increase in quality of life (QoL) was observed in all three groups postoperatively.

**Table 1 jcm-11-07171-t001:** Clinical-Etiological-Anatomical-Pathophysiological (CEAP) score according to patients’ age and type of surgery.

Surgical Technique	Age	CEAP Score					Total
		2	3	4	5	6	
HL&S	<40	10	21	1	0	0	32
	41–55	2	12	2	0	3	19
	>56	0	20	16	5	4	45
Total		12	53	19	5	7	96
Cryostripping	<40	7	24	4	0	0	35
	41–55	0	26	3	0	2	31
	>56	4	23	10	8	2	47
Total		11	73	17	8	4	113
Phlebectomy	<40	4	12	1	0	0	17
	41–55	0	26	1	0	13	40
	>56	5	15	16	7	8	51
Total		9	53	18	7	21	108

**Table 2 jcm-11-07171-t002:** Preoperative and postoperative data.

	HL&S (*n* = 96)	Cryostripping (*n* = 113)	Phlebectomy (*n* = 108)
	CEAP Score 2/3/4/5/6		
	*p*-value/*r*		
Age	*p* = 0.08; *r* = 0.270	*p* = 0.0001; *r* = 0.464	*p* = 0.0019; *r* = 0.296
Gender	*p* = 0.08	*p* = 0.016	*p* = 0.0002
** *QoL—preoperative data* **			
CIVIQ-20	*p* = 0.019; *r* = 0.289	*p* = 0.001; *r* = 0.575	*p* = 0.001; *r* = 0.611
VAS	*p* = 0.049; *r* = 0.263	*p* = 0.005; *r* = 0.348	*p* = 0.049; *r* = 0.286
Eq-5D	*p* = 0.007; *r* = 0.316	*p* = 0.001; *r* = 0.441	*p* = 0.001; *r* = 0.594
PHQ-9	*p* = 0.0451; *r* = 0.189	*p* = 0.009; *r* = 0.335	*p* = 0.001; *r* = 0.424
GAD-7	*p* = 0.011; *r* = 0.125	*p* = 0.124; *r* = 0.158	*p* = 0.233; *r* = 0.116
** *QoL—postoperative data* **			
CIVIQ-20	*p* = 0.001; *r* = 0.352	*p* = 0.001; *r* = 0.615	*p* = 0.001; *r* = 0.635
VAS	*p* = 0.034; *r* = 0.273	*p* = 0.003; *r* = 0.361	*p* = 0.038; *r* = 0.526
Eq-5D	*p* = 0.002; *r* = 0.345	*p* = 0.001; *r* = 0.469	*p* = 0.001; *r* = 0.641
PHQ-9	*p* = 0.013; *r* = 0.232	*p* = 0.004; *r* = 0.354	*p* = 0.001; *r* = 0.438
GAD-7	*p* = 0.001; *r* = 0.162	*p* = 0.045; *r* = 0.281	*p* = 0.014; *r* = 0.235

CEAP: Clinical–etiological–anatomical–pathophysiological clinical class; QoL: quality of life; HL&S: high ligation and stripping; CIVIQ-20: 20-itemChronic Venous Disease quality-of-life Questionnaire; VAS: pain Visual Analog Scale; Eq-5D: EuroQol-5 Dimension; PHQ-9: Patient Health Questionnaire; GAD-7: generalized anxiety disorder; *p* = *p*-value; *p* < 0.05 is statistically significant; *r* = correlation coefficient.

## Data Availability

The data generated in this study may be requested from the corresponding author.

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
