# Peer review of "Measuring the Quality of Life in Patients with Chronic Venous Disease before and Short Term after Surgical Treatment—A Comparison between Different Open Surgical Procedures"

_jcm, 2022, doi:10.3390/jcm11237171_

Round 1
Reviewer 1 Report
I congratulate the authors of this study.
There are some points that need to be clarified as follows:
- Please specify that the study is a prospective study in the method section.
- Please give more details about the exclusion criteria. Did you exclude the patients with established psychiatric diseases or deep vein thrombosis or phlebitis or the patients with post-op complications?
Author Response
Author's Reply to the Review Report (Reviewer 1)
We want to thank you for your suggestions and appreciation. We hope that our response will improve the quality of our paper and will clarify the unclear aspects, too.
- Please specify that the study is a prospective study in the method section.
Our study was a prospective one. We added this information in the main text.
- Please give more details about the exclusion criteria. Did you exclude the patients with established psychiatric diseases or deep vein thrombosis or phlebitis or the patients with post-op complications?
We added more information regarding exclusion criteria. Patients with psychiatric diseases was an exclusion criterion, but we did not encounter any. Instead, patients with other comorbidities or other chronic diseases that may affect QoL was excluded as follow: other cardio-vascular disease (heart failure > NYHA II, atrial fibrillation, documented myocardial infarct, deep vein thrombosis), respiratory disease (asthma, chronic obstructive pulmonary disease), complications of other chronic diseases like diabetes. Oncological patients were not considered as an exclusion criterion because they do not have a surgical indication for CVD treatment until the cure of cancers (if possible). Patients with phlebitis were not excluded.
Reviewer 2 Report
Chronic venous disease is a relatively common condition in industrialized populations. The early stages usually do not disturb the patient's functioning (although they can be a cosmetic and aesthetic problem), but symptomatic ones do. Edema, lipodermatosclerosis, pruritus, and trophic changes cause not only mild discomfort, but can significantly impair the quality of life of patients. Moreover, they can lead to complications. The benefits of the proposed method of treatment should be considered on many levels, not only objective indicators, such as clinical evaluation and color-flow duplex ultrasound (although this is the gold standard), but should also include the assessment of subjective indicators, such as patient satisfaction with treatment and the impact of on his quality of life.
The aim of this study was to evaluate the short-term effects of treatment. In my opinion, 2 weeks is a short time to assess the subjective benefits of the procedure, however, the authors showed that it was enough to show differences. The authors recognized the limitations of the study and demonstrated them in the body of the article. As they rightly noted, it would be worth repeating a similar study in the long-term perspective.
In the title of Table 2 - I propose to specify which pre- and postoperative data it concerns. In addition, the legend below the table should explain what the r and p values mean.
The results are described very generally and focus mainly on the comparison of differences in the quality of life before and after individual procedures. With so many QoL assessment tools, I miss a more detailed reference that would show which aspect of QoL was the most disturbed and which improved the most/least. In my opinion, the work would be more interesting if information was added about what patients were most satisfied with, in which areas of QoL the greatest change occurred.
To sum up, I can say that the work is interesting and deals with a current topic. I believe that after making the addition, it is suitable for publication.
Author Response
Author's Reply to the Review Report (Reviewer 2)
Chronic venous disease is a relatively common condition in industrialized populations. The early stages usually do not disturb the patient's functioning (although they can be a cosmetic and aesthetic problem), but symptomatic ones do. Edema, lipodermatosclerosis, pruritus, and trophic changes cause not only mild discomfort, but can significantly impair the quality of life of patients. Moreover, they can lead to complications. The benefits of the proposed method of treatment should be considered on many levels, not only objective indicators, such as clinical evaluation and color-flow duplex ultrasound (although this is the gold standard), but should also include the assessment of subjective indicators, such as patient satisfaction with treatment and the impact of on his quality of life.
The aim of this study was to evaluate the short-term effects of treatment. In my opinion, 2 weeks is a short time to assess the subjective benefits of the procedure, however, the authors showed that it was enough to show differences. The authors recognized the limitations of the study and demonstrated them in the body of the article. As they rightly noted, it would be worth repeating a similar study in the long-term perspective.
In the title of Table 2 - I propose to specify which pre- and postoperative data it concerns. In addition, the legend below the table should explain what the r and p values mean.
The results are described very generally and focus mainly on the comparison of differences in the quality of life before and after individual procedures. With so many QoL assessment tools, I miss a more detailed reference that would show which aspect of QoL was the most disturbed and which improved the most/least. In my opinion, the work would be more interesting if information was added about what patients were most satisfied with, in which areas of QoL the greatest change occurred.
To sum up, I can say that the work is interesting and deals with a current topic. I believe that after making the addition, it is suitable for publication.
First of all, we would like to thank you for your interest in analysing our manuscript! Your observations highlighted some weak points of this paper and we tried to improve it. So, we bolded the pre- and postoperative dates, and also added in which QoL domain the greatest change occurred. We also made some modifications in Table 2, according to your suggestions. We hope that by following your indications and responding to your queries, the scientific quality of this paper was improved, being a support for the readers understanding, as well.
Reviewer 3 Report
The topic is of interest for doctors dealing with chronic venous insufficiency. Nonetheless, I have the following main concerns:
1. No information is provided on the general characteristics of the study population, except for a few words on age and gender. What about comorbidities, medications, functional status, etc. These are all conditions that may have a significant impact on quality of life and should be taken into account. A table 1 describing the population is needed.
2. It is not clear whether pre-specified criteria were used to decide which surgical procedure had to be used in the individual patients. Why was QoL different between among the three surgical groups? Where then patients different?
3. I have a hard time understanding Table 2. If the authors want to correlate CEAP score and QoL preoperatively and postoperatively, they should put the actual values of CEAP and QoL questionnaires in the Table, and not just the p values.
4. There are important references missing both in the introduction and the discussion, including a Cochrane review.
Author Response
Author's Reply to the Review Report (Reviewer 3)
- No information is provided on the general characteristics of the study population, except for a few words on age and gender. What about comorbidities, medications, functional status, etc. These are all conditions that may have a significant impact on quality of life and should be taken into account. A table 1 describing the population is needed.
Dear reviewer, we would like to thank you for this good observation. Following your query, we observed that we did not present very clear the exclusion criteria in the initial manuscript. Patients with other conditions that may have a significant impact on quality of life were not included in the study, as well as the patients with previous surgeries for varicose veins or patients with psychiatric diseases. We tried to describe in a clearer way the exclusion criteria in the main text of the revised manuscript that we resubmitted, and we hope that now the data are presented in a manner which support the readers understanding.
- It is not clear whether pre-specified criteria were used to decide which surgical procedure had to be used in the individual patients. Why was QoL different between among the three surgical groups? Where then patients different?
The site of the venous reflux was the main criteria used to decide which surgical procedure had to be used in the individual patients.The patients were different. We introduced a paragraph in the main text, Materials and Methods section, to explain this aspect.
- I have a hard time understanding Table 2. If the authors want to correlate CEAP score and QoL preoperatively and postoperatively, they should put the actual values of CEAP and QoL questionnaires in the Table, and not just the p values.
If we were to put the CEAP score values for each type of pre- and postoperative surgery, as well as the outcome of each QoL score, Table 2 would be quite complex. Thus, we decided to present the results clearly and in focus (p=p-value <0.05 is statistically significant; r=correlation coefficient).
- There are important references missing both in the introduction and the discussion, including a Cochrane review.
Thank you for your suggestion, we have added important references, such as Cochrane review.